# Exploiting Contrastive Learning and Numerical Evidence for Confusing Legal Judgment Prediction

**Leilei Gan[1], Baokui Li[1], Kun Kuang[1*], Yating Zhang[2], Lei Wang[3],**
**Luu Anh Tuan[4], Yi Yang[1] and Fei Wu[156*]**

[1]Zhejiang University    [2]Alibaba Group    [3]University of Massachusetts at Amherst
[4]Nanyang Technological University    [5]Shanghai AI Laboratory
[6]Shanghai Institute for Advanced Study of Zhejiang University
{leileigan, libaokui}@zju.edu.cn    ranran.zyt@alibaba-inc.com    anhtuan.luu@ntu.edu.sg

## Abstract

Given the fact description text of a legal case, legal judgment prediction (LJP) aims to predict the case's charge, applicable law article, and term of penalty. A core challenge of LJP is distinguishing between confusing legal cases that exhibit only subtle textual or number differences. To tackle this challenge, in this paper, we present a framework that leverages MoCo-based supervised contrastive learning and weakly supervised numerical evidence for confusing LJP. Firstly, to make the extraction of numerical evidence (the total crime amount) easier, the framework proposes to formalize it as a named entity recognition task. Secondly, the framework introduces the MoCo-based supervised contrastive learning for multi-task LJP and explores the best strategy to construct positive example pairs to benefit all three subtasks of LJP simultaneously. Extensive experiments on real-world datasets show that the proposed method achieves new state-of-the-art results, particularly for confusing legal cases. Additionally, ablation studies demonstrate the effectiveness of each component[1].

## 1 Introduction

Legal judgment prediction (LJP) is one of the most attractive research topics among legal artificial intelligence (Cui et al., 2022; Feng et al., 2022a; Zhong et al., 2020). Given the fact description text of a legal case, LJP aims to predict its charge, applicable law article, and term of penalty (Aletras et al., 2016; Zhong et al., 2018; Yang et al., 2019; Chalkidis et al., 2019; Gan et al., 2021a; Lyu et al., 2022; Liu et al., 2023; Zhang et al., 2023).

One core problem hindering the performance of LJP from being satisfying is confusing legal cases, which have subtle text or number differences, but

---

[1]Our code is available at https://github.com/leileigan/ContrastiveLJP.

[*]Corresponding Author.

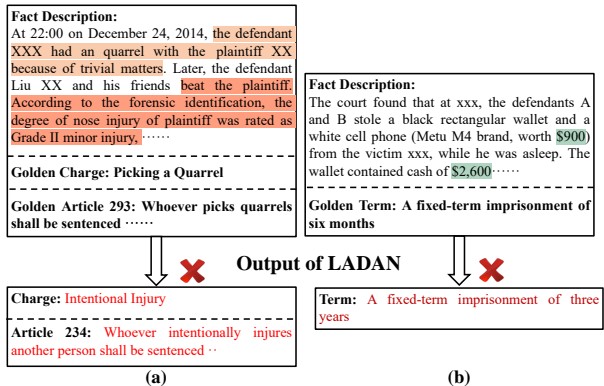

Figure 1: Two examples of confusing legal cases. Figure 1 (a) shows an example whose golden charge label is *Crime of Picking a Quarrel*, which is easily classified into its confusing charge *Crime of Intention Injury*. Figure 1 (b) shows an erroneous prediction of the term of penalty, which does not exploit the amounts related to the crime.

with totally different charges, applicable law articles, or terms of penalty. Figure 1 shows two examples of confusing legal cases. Figure 1 (a) shows an example whose golden charge label is *Crime of Picking a Quarrel*, which is easily classified into its confusing charge *Crime of Intention Injury*. Figure 1 (b) shows an erroneous prediction of the term of penalty, which does not exploit the amounts related to the crime. A series of studies on this problem has been conducted, including manually discriminative legal attributes annotation (Hu et al., 2018), distinguishing representations learning via graph neural networks (Xu et al., 2020), and separating the fact description into different circumstances for different subtasks (Yue et al., 2021).

However, we argue that there exist two drawbacks to these studies. Firstly, all of these studies use a standard cross-entropy classification loss, which cannot distinguish different mistaken classification errors. For example, the error of classifying the charge *Crime of Picking a Quarrel* into its confusing charge *Crime of Intention Injury* is the same

as classifying *Crime of Picking a Quarrel* into a not confusing charge *Crime of Rape*. The model should be punished more if it classifies a charge into its corresponding confusing charges. Secondly, the crime amounts in the fact description are crucial evidence for predicting the penalty terms of certain types of cases, such as financial legal cases. However, the crime amounts are distributed randomly throughout the fact description. Thus it is difficult for the model to directly deduce the precise total crime amount and predict correct penalty terms based on the scattered numbers.

To tackle these issues, we present a framework that leverages numerical evidence and moment contrast-based supervised contrastive learning for confusing LJP. Firstly, the framework proposes to extract numerical evidence (the total crime amount) from the fact description as a named entity recognition (NER) task for predicting the term of the penalty, where the recognized numbers make up the total crime amount. This formulation is able to address the difficulty of directly deducing the precise total crime amount from the fact description where the numbers are scattered randomly throughout and only some of them are part of the crime amount, while others are not. Then, the extracted numerical evidence is infused into the term of penalty prediction model while preserving its numeracy, which is achieved by a pre-trained number encoder model (Sundararaman et al., 2020).

Secondly, in order to pull fact representations from the same class closer and push apart fact representations from confusing charges, the framework introduces the moment contrast-based supervised contrastive learning(SCL; (Khosla et al., 2020; Gunel et al., 2020; Suresh and Ong, 2021)) and explores the best strategy to construct positive example pairs to benefit all three subtasks of LJP simultaneously. The proposed moment contrast-based SCL addresses two challenges to applying the original in-batch SCL to LJP. The first challenge is that the number of charge classes is significantly greater than the number studied in previous studies (Gunel et al., 2020) (e.g., 119 classes for charge prediction), which increases the difficulty of finding sufficient negative examples in the mini-batches. To address it, we introduce a momentum update queue with a large size for SCL, which allows for providing sufficient negative examples. The second challenge is when applying the original single-task SCL to the multi-task LJP there

exists a *Contradictory Phenomenon*. *Contradictory Phenomenon* means that instances with the same charge label may have different applicable law or penalty term labels. If we pair the training instances with the same charge label as positive examples, the resulting learned shared features will benefit the charge prediction task but will degrade the performance of the other two tasks. To tackle this challenge, we explore the best way to construct positive examples, which can benefit all three subtasks of LJP simultaneously.

The proposed framework provides the following merits for predicting confusing LJP: 1) the framework is model-agnostic, which can be used to improve any existing LJP models; 2) the extracted numerical evidence makes the predictions of the term of penalty more interpretable, which is critical for legal judgment prediction; 3) compared with previous studies (Hu et al., 2018), the use of supervised contrastive learning does not require additional manual annotation.

We conduct extensive experiments on two real-world datasets (i.e., CAIL-Small and CAIL-Big). The experimental results demonstrate that the proposed framework achieves new state-of-the-art results, obtaining up to a 1.6 F1 score improvement for confusing legal cases and a 3.73 F1 score improvement for numerically sensitive legal cases. Ablation studies also demonstrate the effectiveness of each component of the framework.

## 2 Related Work

### 2.1 Legal Judgment Prediction

In recent years, with the increasing availability of public benchmark datasets (Xiao et al., 2018a; Feng et al., 2022b) and the development of deep learning, LJP has become one of the hottest topics in legal artificial intelligence (Yang et al., 2019; Zhong et al., 2020; Gan et al., 2021b; Cui et al., 2022; Feng et al., 2022a; Lyu et al., 2022). Our work focuses on confusing legal judgment prediction, which is a typical difficulty in LJP. To solve this challenge, Hu et al. (2018) manually annotates discriminative attributes for legal cases and generates attribute-aware representations for confusing charges by attention mechanism. LADAN (Xu et al., 2020) extracts distinguishable features for law articles by removing similar features between nodes through a graph neural network-based operator. NeurJudge (Yue et al., 2021) utilizes the results of intermediate subtasks to separate the fact de-

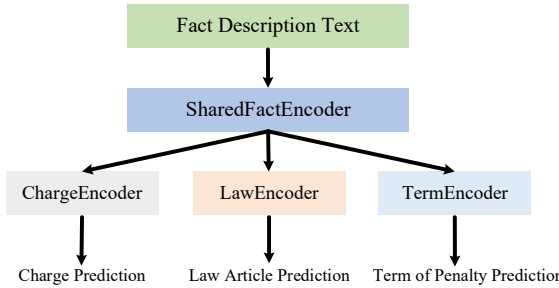

Figure 2: The multi-task learning framework of legal judgment prediction.

scription into different circumstances and exploits them to make the predictions of other subtasks.

## 2.2 Contrastive Learning

The purpose of contrastive learning (Chopra et al., 2005) is to make similar examples closer together and dissimilar examples further apart in the feature space. Contrastive learning has been widely explored for self-supervised/unsupervised representation learning (Wu et al., 2018; Hjelm et al., 2018; Bachman et al., 2019; Chen et al., 2020; He et al., 2020; Nguyen and Luu, 2021; Wu et al., 2022). Recently, several studies have extended contrastive learning to supervised settings (Gunel et al., 2020; Khosla et al., 2020; Suresh and Ong, 2021; Zhang et al., 2022; Nguyen et al., 2022), where examples belonging to the same label in the mini-batch are regarded as positive examples to compute additional contrastive losses. In contrast to previous studies, we present a framework that leverages MoCo-based supervised contrastive learning and numerical evidence, which is neglected by earlier studies for confusing LJP.

## 3 Background

In this section, we formalize the LJP task and its multi-task learning framework.

### 3.1 Problem Formulation

Let $f = \{s_1, s_2, ..., s_N\}$ denote the fact description of a case, where sentence $s_i = \{w_1, w_2, ..., w_M\}$ contains $M$ words, and $N$ is the number of sentences. Given a fact description $f$, the LJP task aims at predicting its charge $y_c \in \mathbb{C}$, applicable law article $y_l \in \mathbb{L}$ and term of penalty $y_t \in \mathbb{T}$.

### 3.2 Multi-Task Learning Framework of LJP

While previous studies have designed various neural architectures for LJP, these models can be boiled down to the following multi-task learning framework as shown in Figure 2. Specifically, firstly, a shared fact encoder is used to encode $f$ into basic legal document representations.

$$\mathbf{H}_f = \text{SharedFactEncoder}(F) \qquad (1)$$

The *SharedFactEncoder* could be Long-Short Term Memory Network (LSTM; (Hochreiter and Schmidhuber, 1997))) or pre-trained langugae models, e.g., BERT (Devlin et al., 2019).

Secondly, in order to learn task-specific representations for each subtask (Zhong et al., 2018; Yue et al., 2021), different private encoders are built upon the basic shared encoder (Zhong et al., 2018; Yue et al., 2021). Specifically, we denote *ChargeEncoder*, *LawEncoder* and *TermEncoder* as corresponding private encoders for the three subtasks as follows.

$$\mathbf{H}_c = \text{ChargeEncoder}(\mathbf{H}_f) \qquad (2)$$
$$\mathbf{H}_l = \text{LawEncoder}(\mathbf{H}_f) \qquad (3)$$
$$\mathbf{H}_t = \text{TermEncoder}(\mathbf{H}_f) \qquad (4)$$

Thirdly, based on these task-specific representations, different classification heads (e.g., multi-layer perceptron) and cross-entropy classification loss are used to compute the losses (i.e., $\ell_c, \ell_l, \ell_t$) for the three tasks. The training objective is the sum of each task's loss as follows:

$$\ell_{ce} = \ell_c + \ell_l + \ell_t \qquad (5)$$

## 4 Methodology

### 4.1 Overview

Fig 3(a) provides an overview of the proposed framework. Given a fact description $f$, on one hand, a well-trained BERT-CRF-based named entity recognition model is used to extract the total crime amount from $f$ as numerical evidence, which is then encoded into representations for predicting the term of the penalty of $f$. On the other hand, a Moco-based supervised contrastive learning for LJP and two strategies for constructing positive example pairs are introduced to compute the contrastive loss. The final training loss is the weighted sum of the contrastive loss and three standard cross-entropy classification losses of subtasks.

### 4.2 Numerical Evidence for Term of Penalty Prediction

**Numerical Evidence Extraction as NER.** In the LJP datasets, there is no explicit total crime amount

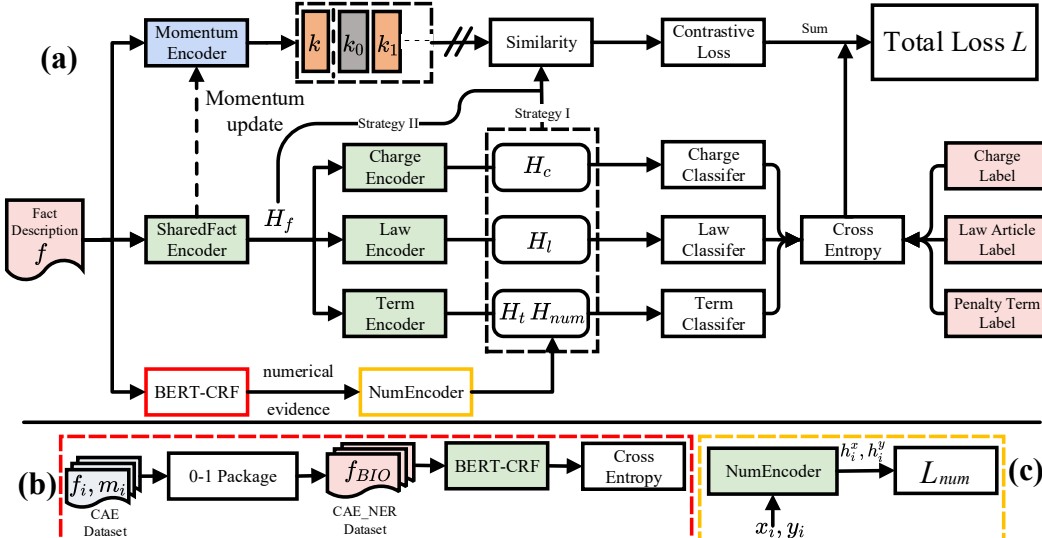

Figure 3: (a) Overview of the proposed framework. (b) Illustration of the training process of the numerical evidence extraction model. (c) Pre-training of the number encoder.

provided for each instance. To address this, we propose to formalize the calculation of the total crime amount as a Named Entity Recognition (NER) task, where the scattered numbers in the fact description that are part of the total crime amount will be recognized as named entities. Then, the sum of the recognized numbers is regarded as the final crime amount. The reason for this formalization is that recognizing which numbers are part of the crime amount is easier than directly computing the total crime amount from the fact description. Fig 3(b) illustrates the training process.

Specifically, we train the numerical extraction model on the dataset used for the Crime Amount Extraction (CAE) task[2]. To train the NER model, given an instance $(f, T)$ in CAE, where $f$ and $T$ denote fact description and crime amount, respectively, we need to convert each instance into the NER format. To reduce expensive manual annotation costs, we propose a 0-1 knapsack algorithm to automatically label named entities in $f$. The 0-1 knapsack algorithm finds a set of sentences from $f$, where the sum of their numbers equals the crime amount $T$. Then the numbers in the selected sentences are labeled as named entities. Algorithm 1 illustrate this construction process. Figure 6 shows an example of converting an instance in CAE into the NER format. The converted dataset is named CAE-NER, based on which we train the state-of-the-art BERT-CRF NER model, referred to as the numerical evidence extraction model.

Now, each instance in the LJP dataset can obtain

[2]http://data.court.gov.cn/pages/laic2021.html

**Algorithm 1:** The 0-1 knapsack algorithm used for automatically constructing the CAE-NER dataset.

**Input** : Fact description $f = \{(s_i, m_i)\}_{i=1}^{N}$, crime amount $T$
**Output** : Selected sentences $\mathbb{E}$

1 **Function** BinarySelect(*f, T, j*):
  /* Select a set of sentences, the sum of their crime amount is $T$  */
2  **for** $i \leftarrow j$ *to* $N$ **do**
3    **if** $m_i < T$ **then**
4      **if** BinarySelect$(f, T - m_i, j + 1)$ **then**
5        $\mathbb{E} = \mathbb{E} \cup \{s_i\}$
6        **return** True
7      **end**
8    **end**
9    **else if** $m_i == T$ **then**
10      $\mathbb{E} = \mathbb{E} \cup \{s_i\}$
11      **return** True
12    **end**
13  **end**
14  **return** $\mathbb{E}$
15 **end**
16 $\mathbb{E} = $ BinarySelect$(f, T, 0)$
17 **return** $\mathbb{E}$

a pseudo crime amount label $m$ annotated by the well-trained numerical evidence extraction model, denoted as a five-tuple $(f, y_c, y_l, y_t, m)$.

**Numerical Evidence Encoder.** Given the extracted numerical evidence, we need a numerical evidence encoder (NumEncoder) that should be capable of encoding the numerical evidence into hidden representations while preserving its numerical significance. To achieve this, we propose to pre-train NumEncoder with the following principle: the

cosine similarity of the learned representations of a pair of numbers should have a linear relationship with respect to their numerical distance. Fig 3(c) illustrates the training process of NumEncoder.

Specifically, given an automatically generated training data $\mathbb{D}_{num} = \{(x_i, y_i)\}_{i=1}^{N}$, where $(x_i, y_i)$ represents a pair of numbers, we use the following training objective $\ell_{num}$ to optimize the parameters of the LSTM-based NumEncoder:

$$\mathbf{x}_i = \text{NumEncoder}(x_i) \tag{6}$$

$$\mathbf{y}_i = \text{NumEncoder}(y_i) \tag{7}$$

$$\ell_{num} = \left\| \frac{2|x_i - y_i|}{|x_i| + |y_i|} - \cos(\mathbf{x}_i, \mathbf{y}_i) \right\| \tag{8}$$

where *cos* represents the cosine distance function.

**Infusing Numerical Evidence for Predicting the Term of Penalty.** Lastly, we infuse the representations of the numerical evidence into the term of the penalty prediction model. Specifically, given a training instance $(f, y_c, y_l, y_t, m)$, its numerical evidence $m$ is encoded by the pre-trained number encoder and then is fused into the term of penalty prediction head as follows:

$$\mathbf{H}_m = \text{NumEncoder}(m) \tag{9}$$

$$\ell_t = \text{CrossEntropy}(\text{MLP}([\mathbf{H}_t; \mathbf{H}_m]), y_t) \tag{10}$$

where $[;]$ denotes the concatenation operation.

## 4.3 MoCo-based Supervised Contrastive Learning for Confusing Judgment Prediction

To address the challenges of large class numbers and the multi-task learning nature of LJP when applying the original in-batch SCL, we introduce the momentum contrast (MoCo) (He et al., 2020) based SCL. Furthermore, we explore the best way to construct positive examples so that they can benefit all three subtasks of LJP simultaneously.

Firstly, we propose to augment the standard in-batch SCL with a large-sized momentum update queue (He et al., 2020), allowing for providing sufficient samples for computing the contrastive loss. Specifically, we maintain one feature queue $\mathcal{Q}$ and one label queue $\mathcal{L}$ to store sample features and corresponding labels. For each example $<e_i, l_i>$ in the mini-batch $\mathcal{I}$, we select positive and negative samples from $\mathcal{Q}$ based on the labels in $\mathcal{L}$ to compute

the supervised contrastive loss as follows:

$$\ell_{sup} = \sum_{i \in \mathcal{I}} -\frac{1}{|P(i)|} \sum_{p \in P(i)} \log \frac{\exp(q_i \cdot k_p / t)}{\sum_{a \in A(i)} \exp(q_i \cdot k_a / t)} \tag{11}$$

where $P(i) = \{t | y_t = y_i, t \in \mathcal{L}\}$ and $A(i) = \{t | t \in \mathcal{L}\}$. $q_i$ is the query feature encoded by a query encoder $f_q(\cdot; \theta_q)$. $k_p, k_a$ in $\mathcal{Q}$ are the key features encoded by a key encoder $f_k(\cdot; \theta_k)$. $\theta_k$ are smoothly updated as follows:

$$\theta_k \leftarrow m\theta_{k-1} + (1-m)\theta_q \tag{12}$$

where $m$ is the momentum coefficient. Samples in $\mathcal{Q}$ and $\mathcal{L}$ are progressively replaced by the current mini-batch following a first-in-first-out strategy. In the ablation section, this MoCo-based SCL shows advantages over the standard in-batch SCL.

Next, we explore two strategies to construct positive example pairs to address the multi-task learning challenge of LJP.

**Strategy I.** A straightforward strategy is to compute a contrastive loss for each subtask of LJP, and then sum them into one loss. Formally, three feature queues, i.e., $\mathcal{Q}^c$, $\mathcal{Q}^l$ and $\mathcal{Q}^t$, are used to store task-specific feature, i.e., $\mathbf{H}_c$, $\mathbf{H}_l$ and $\mathbf{H}_t$. Three label queues, i.e., $\mathcal{L}^c$, $\mathcal{L}^l$ and $\mathcal{L}^t$ are used to store subtask labels, i.e., $y_c$, $y_l$ and $y_t$. The overall contrastive loss is defined as follows:

$$\ell_{cl} = \alpha\ell_{sup}^c + \beta\ell_{sup}^l + \theta\ell_{sup}^t \tag{13}$$

where $\ell_{sup}^c$, $\ell_{sup}^l$ and $\ell_{sup}^t$ are contrastive losses for each subtask computing by Eq. 11. The final training objective of Strategy I is defined by:

$$\ell = \ell_{ce} + \ell_{cl} \tag{14}$$

**Strategy II.** When closely examining Eq. 13, we can observe the *Contradictory Phenomenon* as discussed in Sec. 1. In Strategy I, $\ell_{sup}^{task}$ treats instances with the same subtask labels (e.g., charge labels) as positive examples. However, these instances may have different other subtasks labels (e.g., applicable law labels or term of penalty labels). As a result, $\ell_{sup}^{task}$ will force the $SharedFactEncoder$ to learn features that benefit one subtask but degrade the performance of the other two tasks.

To solve this problem, we propose to view the instances whose three subtask labels are all the same as positive examples and impose the MoCo-based SCL on the shared features $\mathbf{H}_f$. Specifically, we use a feature queue $\mathcal{Q}^B$ to store the shared features $\mathbf{H}_f$, and three label queues $\mathcal{L}^c$, $\mathcal{L}^l$ and $\mathcal{L}^t$ to store three subtask labels. Then the positive

| Dataset | CAIL-Small | CAIL-Big | CAE |
|---|---|---|---|
| #Training Set Cases | 101,619 | 1,588,381 | 3275 |
| #Validation Set Cases | 13,769 | 13,769 | 273 |
| #Test Set Cases | 26,749 | 185,290 | 500 |
| #Charges | 119 | 134 | - |
| #Law Articles | 103 | 121 | - |
| #Term of Penalty | 11 | 11 | - |

Table 1: Statistics of the used legal judgment prediction datasets (CAIL-Small and CAIL-Big) and Crime Amount Extraction (CAE) dataset.

samples set for sample $i$ is denoted as $P(i) = \{q|\mathcal{L}^c(q) = y_i^c, \mathcal{L}^l(q) = y_i^l, \mathcal{L}^t(q) = y_i^t, q \in \mathcal{Q}^B\}$ where $\mathcal{L}^{\text{task}}(q)$ denotes the label of index $q$ in each task label queue. Based on $P(i)$, we can use Eq. 11 to compute the contrastive loss, denote as $\ell_{sup}^B$. Strategy II is able to address the *Contradictory Phenomenon* in Strategy I and improve the performance of all three subtasks.

The final training objective of Strategy II is defined by:

$$\ell = \ell_{ce} + \lambda\ell_{sup}^B \qquad (15)$$

where $\lambda$ is a hyperparameter.

# 5 Experiments

## 5.1 Datasets

To evaluate the effectiveness of our framework, we conduct experiments on two real-world datasets (i.e., CAIL-Small and CAIL-Big) (Xiao et al., 2018b). Each instance in both datasets contains one fact description, one applicable law article, one charge, and one term of penalty. To ensure a fair comparison, we use the code released by (Xu et al., 2020) to process the data. All models are trained on the same dataset. The crime Amount Extraction (CAE) dataset is also a real-world dataset from the Chinese Legal AI challenge [3]. Table 1 shows the statistics of the used datasets.

To specifically evaluate our framework on confusing legal cases, we define a set of confusing and number-sensitive charges. Due to page limitations, the details and statistics of these charges definitions are listed in Table 9 in the Appendix

## 5.2 Implementation Details

We follow (Xu et al., 2020) to conduct data preprocessing. The THULAC [4] tool is used to segment Chinese into words. The word embedding layer in the neural network is initialized by pre-train

[3] http://data.court.gov.cn/pages/laic2021.html
[4] https://github.com/thunlp/THULAC

| Methods | Charges F1 | Articles F1 | Term F1 |
|---|---|---|---|
| LADAN | 82.42 | 75.87 | 34.28 |
| LADAN+I | 82.77 | 76.40 | 34.43 |
| LADAN+II | **83.83** | 77.04 | 35.71 |
| LADAN+I+II | 83.47 | **77.38** | **35.77** |

Table 2: Effects of different supervised contrastive learning strategies.

word embeddings provided by (Zhong et al., 2018). More training details about the BERT-CRF NER model, the LJP model, and the NumberEncoder model can refer to Table 8 in the Appendix.

## 5.3 Baselines

We compare our framework with the following state-of-the-art baseliens: **HARNN** (Yang et al., 2016), **LADAN** (Xu et al., 2020), **NeurJudge+** (Yue et al., 2021), **CrimeBERT** (Zhong et al., 2019). The details of these baselines are left in the Appendix.

## 5.4 Development Experiments

To empirically evaluate which strategy is better for performing SCL for multi-task LJP, we conduct development experiments on CAIL-small using the LADAN backbone. The results are listed in Table 2. As can be observed, firstly, both Strategy I and II lead to improvements in LADAN's performance across the three subtasks. However, the gains of Strategy I is much smaller than those of Strategy II, which verifies the existence of *Contradictory Phenomenon* in Strategy I. Furthermore, we also explore the effect of combining these two strategies, i.e., using $\ell_{cl} + \ell_B$ as the supervised contrastive loss. As seen, the improvement of this combination is not significant.

Consequently, in the remaining experiments, we adopt Strategy II as the contrastive loss, unless otherwise specified. The final method combined the MoCo-based SCL and numerical evidence is denoted as **NumSCL**.

## 5.5 Main Results

To evaluate the effectiveness of the proposed framework, we augment each baseline with **NumSCL** and conduct experiments on the CAIL-Small and CAIL-Big datasets. Due to the expensive training cost and the large size of the training dataset, we did not evaluate CrimeBERT on CAIL-Big following (Yue et al., 2021). The results are listed in Table 3 and Table 4.

| Tasks | Charges | | | | Law Articles | | | | Term of Penalty | | | |
|---|---|---|---|---|---|---|---|---|---|---|---|---|
| Metrics | Acc. | MP | MR | F1 | Acc. | MP | MR | F1 | Acc. | MP | MR | F1 |
| HARNN | 84.54 | 82.56 | 82.94 | 82.26 | 80.09 | 76.46 | 77.69 | 75.95 | 38.38 | 36.12 | 33.99 | 34.32 |
| w/NumSCL | **85.26** | **83.93** | **83.76** | **83.39** | **81.07** | **77.95** | **78.52** | **77.11** | **39.18** | **37.32** | **34.50** | **35.03** |
| LADAN$_{MTL}$ | 84.90 | 82.55 | 83.26 | 82.42 | 80.38 | 75.84 | 77.84 | 75.67 | 38.21 | 35.95 | 34.01 | 34.28 |
| w/NumSCL | **85.37** | **83.91** | **84.04** | **83.57** | **81.32** | **78.06** | **78.59** | **77.24** | **39.38** | **37.95** | **35.23** | **35.95** |
| NeurJudge$^+$ | 83.25 | 82.11 | 81.69 | 81.3 | 80.95 | 77.93 | 78.59 | 77.00 | 37.88 | 37.20 | 33.82 | 34.92 |
| w/NumSCL | **84.45** | **83.30** | **83.55** | **82.88** | **81.12** | **78.10** | **78.98** | **77.32** | **39.65** | **39.48** | **34.65** | **36.22** |
| CrimeBERT | **86.61** | 85.04 | 84.72 | 84.51 | 82.33 | 79.38 | 79.72 | 78.46 | 39.34 | **38.66** | 35.48 | 36.58 |
| w/NumSCL | 85.91 | **85.71** | **85.98** | **85.54** | **82.63** | **80.10** | **80.88** | **79.50** | **39.72** | 38.50 | **35.84** | **36.67** |

Table 3: Main results on CAIL-Small. Acc., MP, and MR are short for accuracy, macro precision, and macro recall, respectively.

| Tasks | Charges | | | | Law Articles | | | | Term of Penalty | | | |
|---|---|---|---|---|---|---|---|---|---|---|---|---|
| Metrics | Acc. | MP | MR | F1 | Acc. | MP | MR | F1 | Acc. | MP | MR | F1 |
| HARNN | 96.48 | 88.10 | 83.54 | 85.34 | 96.54 | 84.88 | 79.42 | 81.40 | **60.30** | 51.08 | 47.47 | 48.79 |
| w/NumSCL | **96.55** | **88.84** | **83.96** | **85.82** | **96.60** | **86.63** | **81.16** | **82.99** | 60.20 | **52.18** | **47.50** | **49.06** |
| LADAN$_{MTL}$ | 96.56 | 88.58 | 84.17 | 85.95 | 96.61 | 86.40 | 80.46 | 82.52 | 60.44 | 51.56 | 48.67 | 49.78 |
| w/NumSCL | **96.65** | **89.34** | **84.57** | **86.49** | **96.71** | **87.60** | **81.74** | **83.68** | **60.56** | **51.85** | **48.86** | **49.81** |
| NeurJudge$^+$ | 95.48 | 85.57 | 79.55 | 81.49 | **96.26** | **85.78** | 81.38 | **82.80** | 58.40 | **49.67** | 43.32 | 44.90 |
| w/NumSCL | **95.73** | **86.37** | **80.88** | **82.58** | 96.20 | 85.48 | **81.56** | 82.78 | 58.40 | 49.10 | **43.54** | **45.43** |

Table 4: Main results on CAIL-Big. Acc., MP and MR are short for accuracy, macro precision, and macro recall, respectively.

From Table 3 and Table 4, we make the following observations. Firstly, the proposed framework can improve all the baselines and achieve new state-of-the-art results on the two datasets. Specifically, on CAIL-Small, the absolute improvements reach up to 1.15, 1.57, and 1.67 F1 scores for the charge, law article, and term of penalty predictions, respectively. Secondly, on CAIL-Big, the gains are smaller, giving absolute improvements of 0.54, 1.16, and 0.53 F1 scores for the charge, law article, and term of penalty predictions, respectively. Thirdly, we observe that on CAIL-Big, NeurJudge$^+$ gives worse performance than the other baselines. We hypothesize that the complex neural network architecture of NeurJudge$^+$ may lead to overfitting on the CAIL-Big dataset. Lastly, for CrimeBERT, our framework can still obtain an absolute improvement of 1.03 and 1.04 F1 scores for charge and law article predictions. The overall gain on the term of penalty prediction is slight, however, when specifically evaluating number-sensitive legal cases, the improvement can still be up to a 1.79 F1 score, as shown in Table 5.

## 5.6 Ablation Studies

**Effect of Numerical Evidence for Number-Sensitive Legal Cases.** To examine the effect of numerical evidence for predicting the term of penalty, we conduct ablative experiments. As

| Models | Acc. | F1 | Num. F1 |
|---|---|---|---|
| HARNN | 38.38 | 34.32 | 28.13 |
| w/ SCL | 38.62 | 34.45 | 28.24(↑ 0.11) |
| w/ NumSCL | 39.18 | 35.03 | 29.15(↑ 1.02) |
| LADAN$_{MTL}$ | 38.21 | 34.28 | 26.83 |
| w/ SCL | 39.50 | 35.71 | 28.54(↑ 1.71) |
| w/ NumSCL | 39.38 | 35.95 | 30.56(↑ 3.73) |
| NeurJudge$^+$ | 37.88 | 34.92 | 27.30 |
| w/ SCL | 38.11 | 34.96 | 27.35(↑ 0.05) |
| w/ NumSCL | 39.65 | 36.22 | 29.03(↑ 1.73) |
| CrimeBERT | 39.34 | 36.58 | 28.57 |
| w/ SCL | 39.71 | 36.48 | 28.99(↑ 0.42) |
| w/ NumSCL | 39.68 | 36.66 | 30.36(↑ 1.79) |

Table 5: Effects of the proposed method on the term of penalty prediction of number-sensitive legal cases. Num. F1 is the F1 score on the defined number-sensitive charges.

shown in Table 5, the improvement of MoCo-based SCL only is relatively tiny, only giving 0.11 and 0.05 F1 score improvements for HARNN and NeurJudge$^+$ on number-sensitive legal cases. However, when the models are further provided with the extracted numerical evidence, the F1 scores of all the baselines have a considerable boost. In particular, LADAN$_{MTL}$ obtains a 3.73 F1 score improvement on number-sensitive cases. These results show that the extracted crime amount is more beneficial for number-sensitive legal cases.

**Effect of Contrastive Learning for Confusing Charges.** We conduct experiments to validate the

| Models | Acc. | F1 | Conf. F1 |
|---|---|---|---|
| HARNN | 84.54 | 82.26 | 75.60 |
| w/ NumSCL | 85.26 | 83.39 | 77.19 (↑ 1.59) |
| LADAN$_{MTL}$ | 84.90 | 82.42 | 75.29 |
| w/ NumSCL | 85.37 | 83.57 | 76.76 (↑ 1.47) |
| NeurJudge$^+$ | 83.25 | 81.30 | 73.42 |
| w/ NumSCL | 84.45 | 82.88 | 75.97(↑2.55) |

Table 6: Effects of the proposed method on the charge prediction of confusing legal cases. Conf. F1 is the F1 score on the defined confusing charges.

| Models | Charge F1 | Law F1 | Term F1 |
|---|---|---|---|
| LADAN | 82.42 | 75.67 | 34.28 |
| LADAN$_{SCL}$ | 83.18 | 76.24 | **36.12** |
| LADAN$_{NumSCL}$ | **83.57** | **77.24** | 35.95 |

Table 7: Effect of momentum contrast queue.

effect of contrastive learning for predicting charges, particularly confusing charges. As shown in Table 6, the absolute F1 score improvements of confusing charges are greater than those of the overall charges. For example, NeurJudge$^+$ obtains an absolute 2.55 F1 score improvement on confusing charges, which demonstrates the effectiveness of the moco-based SCL in learning distinguishable representations for confusing charges.

**Effect of Momentum Contrast Queue.** We utilize LADAN as the backbone for comparing NumSCL with the in-batch SCL(SCL) which takes the current mini-batch as the lookup dictionary to compute the contrastive loss. The in-batch SCL is trained using the same parameters as the MoCo-based SCL. As depicted in Table 7, the in-batch SCL yields superior results when combined with LADAN but underperforms NumSCL in charge and law article prediction tasks. This highlights the advantage of employing a large queue as the lookup dictionary in SCL. We also observe that the performance of the term of penalty is not significantly affected by the choice of the lookup dictionary. This observation aligns with the previous finding that the extracted numerical evidence is more beneficial than SCL for the term of penalty prediction.

**Effect of $\lambda$.** We carry out experiments to verify the impact of $\lambda$ in Eq.15. As illustrated in Figure 5, with the increasing of $\lambda$, the performance of charge and law article predictions correspondingly improves. We also observe a fluctuation in the term of penalty prediction, again showing that the extracted numerical evidence plays a more significant role than SCL for predicting the term of penalty.

| Example 1 |
|---|
| **Fact Descriptions:** At XXX, the defendant XXX fought with the victim XXX due to trivial matters in a market in XXX Town, XXX City, and wanted to take revenge on the victm XXX. At XXX of the same year, the defendant XXX gathered three people including XXX, XXX and XXX, and fled to the open-air parking lot next to a market. The defendant XXX slashed the victim XXX's right arm with a watermelon knife, then they fled the scene. According to the forensic identification, the victim XXX suffered minor injuries⋯⋯ |
| **Golden Charge:** Crime of Picking a Quarrel; **Goden Law Article:** 34; **Golden Term of Penalty:** 6 |
| **LADAN$_{MTL}$:** Crime of Intention Injury; Law Article 34; Term of Penalty: 4 |
| **LADAN$_{MTL}$+NumSCL:** Crime of Picking a Quarrel; Law Article 34; Term of Penalty: 6 |
| Example 2 |
| **Fact Descriptions:** From XXX to XXX, the defendant XXX committed three thefts in XXX City. 1. At XXX, 2015, the defendant XXX stole a "Dayang brand" electric tricycle worth $5,600 from the victim XXX. 2. On XXX, 2015, the defendant XXX stole a "Dayang" electric tricycle worth $4,800 from the victim XXX in XXX city. 3. At XXX, 2016, the defendant XXX stole a electric vehicle battery from the victim XXX, but was discovered and attempted. |
| **Golden Charge:** Crime of Theft; **Golden Law Article** 32; **Golden Term of Penalty**: 7 |
| **LADAN$_{MTL}$:** Crime of Theft; Law Article 56; Term of Penalty:9 |
| **LADAN$_{MTL}$+NumSCL:** Crime of Theft; Law Article 56; Term of Penalty: 7; Extracted Crime Amount: $10,400 |

Figure 4: Qualitative examples to demonstrate the effect of the proposed framework.

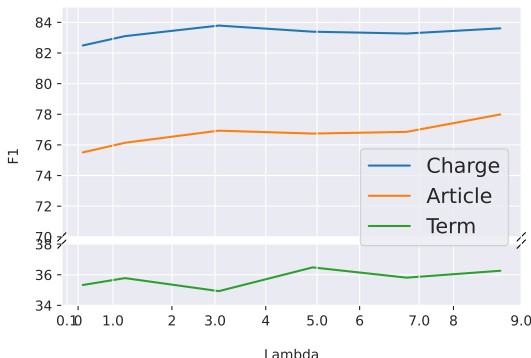

Figure 5: The impact of $\lambda$ in Eq.15 for the three subtasks of LJP.

## 6 Case Studies

Figure 4 shows two cases to qualitatively demonstrate the effect of the proposed framework. In the first case, LADAN$_{MTL}$ incorrectly predicts the case's charge into its confusing charge *Crime of Intentional Injury*, which should be *Crime of Picking a Quarrel*. However, with the proposed framework, this error is corrected. The second case demonstrates the effect of numerical evidence. LADAN$_{MTL}$ incorrectly predicts the case's term of penalty as label 9, meaning a sentence of fewer than 6 months. Given the accurately extracted crime amount of $10,400, which is a relatively large crime amount, the model correctly predicts

the term of penalty as label 7, meaning a sentence of more than 9 months but less than 12 months.

## 7 Conclusion

In this paper, we present a framework that introduces MoCo-based supervised contrastive learning and weakly supervised numerical evidence for confusing legal judgment prediction. The framework is capable of automatically extracting numerical evidence for predicting number-sensitive cases and learning distinguishable representations to benefit all three subtasks of LJP simultaneously. Extensive experiments validate the effect of the framework.

## Limitations

While the used 0-1 knapsack algorithm enjoys the merit of automatically constructing a training dataset for building the NER model, it cannot accurately calculate the crime amount when the suspects return some properties to the victims as the returned properties should be subtracted from the amount of the crime. More sophisticated techniques could be developed to calculate the amount of crime more precisely.

Our LJP research focuses on Chinese legal documents under the jurisdiction of the People's Republic of China. While the framework was developed and tested specifically for the 3-task Chinese Legal Judgment Prediction (LJP), we believe the underlying methodology could be generalized and applied to other LJP tasks, even those from different jurisdictions. However, this would likely require modifications to account for the unique characteristics and complexities of each jurisdiction's legal system. We will leave this for future work.

## Ethical Concerns

Due to the sensitive nature of the legal domain, applying artificial intelligence technology to the legal field should be carefully treated. In order to alleviate ethical concerns, we undertake the following initiatives. First, to prevent the risk of leaking personal private information from the evaluated real-world datasets, sensitive information, such as names of individuals and locations, has been anonymized. Second, we suggest the predictions generated by our model should serve as supportive references to assist judges in making judgments more efficiently, rather than solely determining the judgments.

## Acknowledgements

This work was supported in part by National Key Research and Development Program of China (2022YFC3340900), National Natural Science Foundation of China (62376243, 62037001), the StarryNight Science Fund of Zhejiang University Shanghai Institute for Advanced Study (SN-ZJU-SIAS-0010), Project by Shanghai AI Laboratory (P22KS00111), Program of Zhejiang Province Science and Technology (2022C01044), the Fundamental Research Funds for the Central Universities (226-2022-00142, 226-2022-00051).

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

## A  Baselines

We compare our method with the following non-pretrained and pre-trained models: (1) **HARNN** (Yang et al., 2016): an RNN-based neural network with a hierarchical attention mechanism for document classification; (2) **LADAN** (Xu et al., 2020): LADAN distinguishes confusing law articles by extracting distinguishable features from similar law articles using a graph-based method; (3) **NeurJudge+** (Yue et al., 2021):NeurJudge+ utilizes the results of intermediate subtasks to separate

| Model | Parameter | Value |
|---|---|---|
| LJP | Word embedding size | 200 |
| | Maximum document length | 512 |
| | Maximum sentence num | 15 |
| | Batch size | 128 |
| | Learning rate | 0.001 |
| | Training epoch | 16 |
| | Optimizer | Adam |
| | Momentum queue $\mathcal{Q}$ size | 65536 |
| | Momentum coefficient $m$ | 0.999 |
| | Temperature t | 0.07 |
| | $\alpha, \beta, \theta, \lambda$ | 2, 2, 5, 7 |
| NER | Model | BERT-CRF |
| | Batch size | 16 |
| | Learning rate | 0.00001 |
| | Training epoch | 20 |
| | Optimizer | Adam |
| NumEncoder | Training data size | 128000 |
| | Max number, Min number | 0, 300000 |
| | Model | GRU |
| | Word embedding size | 200 |
| | Hidden state size | 256 |
| | Batch size | 128 |
| | Learning rate | 0.001 |
| | Training epoch | 100 |
| | Optimizer | Adam |

Table 8: Hyper-parameter values

the fact description into different representations and exploits them to make the predictions of other subtasks; (4) **CrimeBERT** (Zhong et al., 2019): CrimeBERT is initialized by BERT (Devlin et al., 2019), then is further pre-trained on crime data, giving better results than BERT. It is worth noting that the proposed method is model-agnostic, which can be used to improve any other LJP models.

## B  Implementation Details

For training the BERT-CRF NER model, we use the Adam optimizer and set the learning rate to 1e-5. The batch size is set to 16. We train the model for 20 epochs and select the best model on the validation set for testing. The best model on the test set of CAE can achieve a competitive accuracy of 0.875.

For pre-training the numerical evidence encoder model NumEncoder, we synthesize a training dataset of size 128,000 where each pair of numbers $(x_i, y_i)$ are uniformly sampled from $[0, 30, 0000]$. A GRU model with a hidden size of 256 is used to model the number sequence. Adam is used to optimizing the parameters, and the learning rate is set to 1e-3. We train the NumEncoder model for 100 epochs.

For training the LJP model, we use the Adam optimizer and set the learning rate to 1e-3. The

**Fact Description:**

The court found that at XXX, the defendant XXX stole a black rectangular wallet and a white cell phone (Meitu M4 brand, worth $900^MONEY^) from the victim XXX, while he was asleep. The wallet contained cash of $2600^MONEY^. After Being arrested, the defendant returned the aforementioned cell phone and cash of $2,000 to the victim.

- - - - - - - - - - - - - - -

**Crime Amount: $3,500**

Figure 6: An example of converting an instance in the CAE dataset into the sample for training the NER model.

| Charge Type | #Classes | %Training Set size |
|---|---|---|
| Confusing Charges | 41 | 37.13% |
| Number-sensitive Charges | 33 | 22.25% |

Table 9: Statistics of the defined confusing charges and number-sensitive charges.

batch size is set to 128. We train the model for 16 epochs and select the best model on the validation set for testing. In contrastive learning, the MoCo queue size and the temperate $t$ are set to 65536 and 0.07, respectively. We run each experiment with five different seeds and report the averaged results.

Table 1 shows the detailed statistics of the used datasets.

## C  Evidence Extraction

Figure 6 shows an example of converting an instance in CAE into the NER format.

## D  Definition of Confusing Charges.

To specifically evaluate our method on confusing legal cases, we define confusing charges using the predicted results of the baseline model LADAN$_{MTL}$. Concretely, if the number that the model incorrectly classifies class A into class B exceeds the pre-defined threshold, classes A and B will be added to the confusing classes. The definition of the number-sensitive charges is determined by an experienced legal expert. The statistics of the definition of confusing charges and number-sensitive charges are listed in Table 9.