# OpenReview forum: "Exploiting Contrastive Learning and Numerical Evidence for Confusing Legal Judgment Prediction"
_EMNLP/2023/Conference — EMNLP 2023 Findings_

### Official Review · Reviewer_VZ2V · 2023-08-01

**Soundness:** 4

**Excitement:**

3: Ambivalent: It has merits (e.g., it reports state-of-the-art results, the idea is nice), but there are key weaknesses (e.g., it describes incremental work), and it can significantly benefit from another round of revision. However, I won't object to accepting it if my co-reviewers champion it.

**Paper Topic And Main Contributions:**

The authors propose a new framework for Legal Judgment Prediction (LJP) for the Chinese CAIL multi-task datasets, where a model has to predict the relevant criminal charge, law articles, and penalty term based on criminal case facts. The authors propose (a) the use of contrastive learning to better discriminate charges, and (b) the use of a numerical evidence extractor to enhance the penalty term predictor.
The authors experiment with two datasets (CAIL-Small, and CAIL-Big), and compare the use of the framework to several other baselines (HARNN, LADAN, NeurJudge+, CrimeBERT). The absolute performance improvement for using the proposed framework is approx. 1-1.5% across all methods.

**Questions For The Authors:**

- The authors mention that using standard cross-entropy is not an ideal solution, since "The model should be punished more if it classifies a charge into its corresponding confusing charges.". This sounds counter-intuitive to me since misclassifications naturally occur in closed-related subjects (close-related labels, e.g., similar types of crimes). I don't understand why such misclassifications should be penalized more compared to misclassifications in favor of very unrelated crimes. Intuitively, misclassifying the crime of "Picking a Quarrel" as "Intentional Injury" sounds less problematic compared to picking a much more unrelated crime such as "Drug Trafficking" for instance. Can you explain the reasoning (intuition) behind the proposed solution?

**Reasons To Accept:**

- The authors propose an interesting framework relying on contrastive learning to "push" the predictive performance in the Chinese LJP multi-task benchmarks.

**Reasons To Reject:**

-  The proposed framework is built around a specific setting (3-task Chinese LJP), and there is no evidence that a similar methodology would be useful in other related LJP tasks from other jurisdictions (Chalkidis et al., 2019; Niklaus et al., 2021; and others).

**Reproducibility:**

3: Could reproduce the results with some difficulty. The settings of parameters are underspecified or subjectively determined; the training/evaluation data are not widely available.

**Reviewer Confidence:**

1: Not my area, or paper was hard for me to understand. My evaluation is just an educated guess.

**Typos Grammar Style And Presentation Improvements:**

- The authors should make clear and mention that they use datasets for LJP in the Chinese language in the context of People's Republic of China (PRC) jursdiction (law). There are phrases like "LJP aims to predict its charge, applicable law article, and term of penalty.", which sound like the LJP is a universal task across languages and countries. The Bender Rule: "Always name the language you're working on" is not only applicable to English.

---

> ### Author Rebuttal · Authors · 2023-08-29
>
> Q1: The proposed framework is built around a specific setting (3-task Chinese LJP), and there is no evidence that a similar methodology would be useful in other related LJP tasks from other jurisdictions (Chalkidis et al., 2019; Niklaus et al., 2021; and others).
>
> A1: Thank you for bringing up this important point. While our framework was developed and tested specifically for the 3-task Chinese Legal Judgment Prediction (LJP), we believe the underlying methodology could be generalized and applied to other LJP tasks, even those from different jurisdictions. However, this would likely require modifications to account for the unique characteristics and complexities of each jurisdiction's legal system. We will leave this for future work.
>
> Q2: The authors mention that using standard cross-entropy is not an ideal solution, since "The model should be punished more if it classifies a charge into its corresponding confusing charges.".
>
> A2:  Thank you for your valuable comments. We found that standard cross-entropy classification loss struggles to distinguish confusing charges, therefore we propose an additional contrastive learning loss to punish the model more if it classifies a charge into its corresponding confusing charges. In other words, the total loss will increase if the model classifies a charge into its corresponding confusing charges. In this way, the model can learn to avoid such errors. This approach encourages the model to pay closer attention to the subtle differences between closely related charges, thereby improving its overall accuracy and performance.
>
> Q3: The authors should make clear and mention that they use datasets for LJP in the Chinese language in the context of the People's Republic of China (PRC) jurisdiction (law).
>
> A3: Thank you for your valuable comments. We will update the manuscript to state upfront that our LJP research focuses on Chinese legal documents under the jurisdiction of the People's Republic of China. Phrases like "LJP aims to predict..." will be rephrased to indicate we are referring specifically to the Chinese LJP task, rather than implying it is universally applicable.

---

### Official Review · Reviewer_kzgp · 2023-08-04

**Soundness:** 3

**Excitement:**

3: Ambivalent: It has merits (e.g., it reports state-of-the-art results, the idea is nice), but there are key weaknesses (e.g., it describes incremental work), and it can significantly benefit from another round of revision. However, I won't object to accepting it if my co-reviewers champion it.

**Missing References:**

[1] Augmenting Legal Judgment Prediction with Contrastive Case Relations
[2] Contrastive Learning for Legal Judgment Prediction
[3] Improving legal judgment prediction through reinforced criminal element extraction

**Paper Topic And Main Contributions:**

This paper presents a novel framework, termed MoCo-based Supervised Contrastive Learning for Multi-task Legal Judgement Prediction (LJP), designed to alleviate the complexity inherent in deciphering legal cases. The approach incorporates numerical evidence extraction, addressing the challenge of predicting penalty terms based on dispersed numerical data. Empirical results indicate a performance enhancement over existing methodologies.

**Reasons To Accept:**

1. The proposed methodology capitalizes on the numerical data embedded within LJP texts, demonstrating a notable improvement in performance.

2. The authors innovatively apply a momentum contrast-based supervised contrastive learning approach, addressing the issue of large class numbers and the subsequent difficulty in identifying sufficient negative examples within mini-batches.

3. The introduced framework exhibits versatility, being adaptable to current encoder-based methodologies.

**Reasons To Reject:**

1.	The paper lacks a comprehensive review and comparison with existing LJP methods [1,2, 3].
2.	The Supervised Contrastive Learning (SCL) methods proposed within this research did not demonstrate a significant enhancement in the ablation studies. This calls for further investigation and optimization.


**Reproducibility:**

4: Could mostly reproduce the results, but there may be some variation because of sample variance or minor variations in their interpretation of the protocol or method.

**Reviewer Confidence:**

5: Positive that my evaluation is correct. I read the paper very carefully and I am very familiar with related work.

---

> ### Author Rebuttal · Authors · 2023-08-29
>
> Q1: The paper lacks a comprehensive review and comparison with existing LJP methods [1,2, 3].
>
> A1: Thank you for your valuable comments and suggestions. Here we give a review and comparison of these three methods.
>
> [1] proposes a relational attention module to impose a relational constraint on the obtained <similar case, case, dissimilar case> to more effectively utilize the beneficial information contained in the data. Different from [1],  our framework leverages MoCo-based supervised contrastive learning and weakly supervised numerical evidence for confusing LJP.
>
> [2] proposes a supervised contrastive learning framework to distinguish various law articles within the same chapter of a Law and similar charges of the same law article or related law articles.  Our work has two key differences compared with [1]. Firstly, we use a moco-based supervised contrastive learning which can provide sufficient negative examples for computing contrastive loss in LJP. Secondly, in addition to supervised contrastive learning, we also propose to extract numerical evidence (the total crime amount) from the fact description for predicting the term of the penalty.  The effect of numerical evidence can be found in Table 5 of our work and Table 3 of [2], which shows that only using supervised contrastive learning does not significantly improve the performance of term of penalty prediction.
>
> [3] proposes a reinforcement learning (RL) based framework to extract four types of criminal elements (i.e., the criminal, target, intentionality, and criminal behavior) to distinguish similar fact descriptions. Different from [3], our work proposes to extract numerical evidence (the total crime amount) from the fact description as a named entity recognition (NER) task for predicting the term of the penalty.
>
> We will include and discuss these related papers in our revised version.
>
> Q2: The Supervised Contrastive Learning (SCL) methods proposed within this research did not demonstrate a significant enhancement in the ablation studies. This calls for further investigation and optimization.
>
> A2: The SCL method is more beneficial for confusing charge legal cases than number-sensitive legal cases. As shown in Table 6, the absolute F1 score improvements of confusing charges are greater than those of the overall charges. However, when conducting ablative experiments to examine the effect of numerical evidence for predicting the term of penalty, as shown in Table 5, the improvement of MoCo-based SCL only is relatively tiny.  When the models are further provided with the extracted numerical evidence, the F1 scores of all the baselines have a considerable boost. These results show that the extracted crime amount is more beneficial for number-sensitive legal cases.

---

### Official Review · Reviewer_zrQE · 2023-08-08

**Soundness:** 3

**Excitement:**

4: Strong: This paper deepens the understanding of some phenomenon or lowers the barriers to an existing research direction.

**Paper Topic And Main Contributions:**

Legal Judgment Prediction (LJP) involves forecasting the charge, relevant law article, and penalty duration for a legal case based on its factual description. LJP encounters a central obstacle for differentiating the legal cases that display subtle variations in text or numeric values.
To authors propose a MoCo-based supervised contrastive learning and weakly supervised numerical cues for classifying LJP scenarios.
They suggest the novel approach of using named entity recognition task to enhance the identification of numerical cues.
Their proposed framework also incorporates MoCo-based supervised contrastive learning to facilitate multi-task LJP and investigates optimal strategies for forming positive example pairs that concurrently benefit all three LJP subtasks.
They try their proposed methods on real world data and compare their final framework with ablated frameworks and with baselines.

**Reasons To Accept:**

The paper is well-written and understandable.
The problem of confusing legal judgment prediction is very important and relevant to the conference.
Their novelty for extraction of numerical evidence as a named entity recognition task is very neat.

**Reasons To Reject:**

Major:
There are no confidence intervals reported for the results in tables 3, 4, 5, 6, and 7. Must repeat the experiments for different random seeds.
The improvements in performance are mostly very small and it is not clear if those small improvements are statistically significant.

Minor:
The momentum contrast (MoCo) is not explained comprehensively. It is very difficult to find its concrete definition in the paper.

**Reproducibility:**

4: Could mostly reproduce the results, but there may be some variation because of sample variance or minor variations in their interpretation of the protocol or method.

**Reviewer Confidence:**

3: Pretty sure, but there's a chance I missed something. Although I have a good feel for this area in general, I did not carefully check the paper's details, e.g., the math, experimental design, or novelty.

---

> ### Author Rebuttal · Authors · 2023-08-29
>
> Q1: There are no confidence intervals reported for the results in Tables 3, 4, 5, 6, and 7.
>
> A1: Thank you for your valuable comments. For the CAIL-Small dataset, as indicated in Lines 763 - 764, I would like to clarify that five different random seeds (10, 512, 1024, 2020, 9090) were used to initialize the weights of the models and shuffle the input data to ensure the robustness and reliability of the experimental results. Therefore, the results in Table 3 are averaged results. The results with confidence intervals of Table 3 are reported as follows:
>
> | Metrics    | Charge Acc | Charge MP | Charge MR | Charge F1 | Law Acc | Law MP | Law MR | Law F1 | Term Acc | Term MP | Term MR | Term F1 |
> |-----------|---------|---------|---------|---------|---------|---------|---------|---------|---------|----------|----------|----------|
> | HARNN     | 84.54±0.56 | 82.56±0.14 | 82.94±0.19 | 82.26±0.05 | 80.09±2.63 | 76.46±0.59 | 77.69±0.26 | 75.95±0.52 | 38.38±0.47 | 36.12±0.91 | 33.99±1.75 | 34.32±0.50 |
> | w/NumSCL  | 85.26±0.14 | 83.93±0.12 | 83.76±0.10 | 83.39±0.09 | 81.07±1.35 | 77.95±1.48 | 78.52±0.69 | 77.11±1.42 | 39.18±0.20 | 37.32±0.61 | 34.50±0.85 | 35.03±0.14 |
> | LADAN_MTL | 84.9±0.42  | 82.55±0.24 | 83.26±0.13 | 82.42±0.20 | 80.38±0.16 | 75.84±0.32 | 77.84±0.24 | 75.67±0.27 | 38.21±0.17 | 35.95±0.81 | 34.01±0.78 | 34.28±0.03 |
> | w/NumSCL  | 85.37±0.21 | 83.91±0.27 | 84.04±0.13 | 83.57±0.20 | 81.32±0.01 | 78.06±0.48 | 78.59±0.21 | 77.24±0.26 | 39.38±0.21 | 37.95±1.15 | 35.23±0.62 | 35.95±0.29 |
> | NeurJudge+| 83.25±0.32 | 82.11±0.09 | 81.69±0.41 | 81.30±0.17 | 80.95±0.11 | 77.93±0.34 | 78.59±0.55 | 77.00±0.46 | 37.88±0.66 | 37.20±0.56 | 33.82±0.41 | 34.92±0.38 |
> | w/NumSCL  | 84.45±0.39 | 83.3±0.29  | 83.55±0.17 | 82.88±0.13 | 81.12±0.68 | 78.10±0.79 | 78.98±0.29 | 77.32±0.47 | 39.65±1.09 | 39.48±0.52 | 34.65±1.04 | 36.22±1.07 |
> | CrimeBERT | 86.61±0.10 | 85.04±0.27 | 84.72±0.08 | 84.51±0.17 | 82.33±0.14 | 79.38±0.72 | 79.72±0.20 | 78.46±0.36 | 39.34±0.42 | 38.66±1.02 | 35.48±0.14 | 36.58±0.22 |
> | w/NumSCL  | 86.91±0.10 | 85.71±0.05 | 85.98±0.06 | 85.54±0.01 | 82.63±1.40 | 80.10±0.09 | 80.88±0.11 | 79.50±0.16 | 39.72±0.19 | 38.50±0.50 | 35.84±0.14 | 36.67±0.23 |
>
> Table 3: Main results on CAIL-Small. Acc., MP, and MR are short for accuracy, macro precision, and macro recall, respectively. We run the experiments with five different seeds and report the averaged results and confidence intervals.
>
> For the CAIL-Big dataset, in our preliminary experiment, we found that due to the large dataset size, experimental results are less affected by the random seed. Therefore, we randomly choose a seed and use this seed for all the experiments.
>
> We will update these tables with confidence intervals in the revised version.
>
> Q2: The momentum contrast (MoCo) is not explained comprehensively. It is very difficult to find its concrete definition in the paper.
>
> A2: Thank you for your valuable comments. Here we illustrate the MoCo-based supervised contrastive learning again.
>
> We propose to augment the standard single-task supervised contrastive learning with a momentum update queue~\cite{he2020momentum}. The queue size can be much larger than a typical mini-batch size, thus can provide sufficient samples for computing the contrastive loss. Specifically, we maintain one feature queue $\mathcal{Q}$ to store sample features of interest, and one label queue $\mathcal{L}$ to store corresponding sample labels. Given $\mathcal{Q}$ and $\mathcal{L}$, for each example <$e_i$, $l_i$> in the mini-batch $\mathcal{I}$, according to the labels in $\mathcal{L}$, we can select positive and negative samples from $\mathcal{Q}$ to compute the supervised contrastive loss using Eq. (11).
>
> In Eq. (11), $q_i$ is the query feature encoded by a query encoder $f_q$ whose parameters are denoted as $\theta_q$. The features $k_p$, $k_a$ in $\mathcal{Q}$ are encoded by a momentum key encoder $f_k$ whose parameters are $\theta_k$. $\mathcal{Q}$, $\mathcal{L}$, and $\theta_k$ are updated as follows:
> $$
> \theta_{k} \leftarrow m\theta_{k-1} + (1-m)\theta_q
> $$
>
> $$
> \text{Dequeue}(\mathcal{Q}), \text{Enqueue}(\mathcal{Q}, k_i)
> $$
>
> $$
> \text{Dequeue}(\mathcal{L}), \text{Enqueue}(\mathcal{L}, l_i)
> $$
>
> where $m \in [0, 1)$ is a momentum coefficient. $\textit{Dequeue}$ and $\textit{Enqueue}$ are operations to remove the element at the front of the queue and insert the element at the end of the queue, respectively. $k_i = f_k(x_i;\theta_k)$. Note that $\theta_k$ are fixed during the back-propagation.

---

### Meta-Review · Area_Chair_S82W · 2023-09-12

**Recommendation:** 3

**Metareview:**

The proposed methodology capitalizes on the numerical data embedded within LJP texts, demonstrating a notable improvement in performance.
The authors innovatively apply a momentum contrast-based supervised contrastive learning approach, addressing the issue of large class numbers and the subsequent difficulty in identifying sufficient negative examples within mini-batches.
The introduced framework exhibits versatility, being adaptable to current encoder-based methodologies.

The proposed framework is built around a specific setting (3-task Chinese LJP), and there is no evidence that a similar methodology would be useful in other related LJP tasks from other jurisdictions (Chalkidis et al., 2019; Niklaus et al., 2021; and others).

---

### Decision · Program_Chairs · 2023-10-07

**Decision:**

Accept-Findings

**Comment:**

The proposed methodology capitalizes on the numerical data embedded within LJP texts, demonstrating a notable improvement in performance.
The authors innovatively apply a momentum contrast-based supervised contrastive learning approach, addressing the issue of large class numbers and the subsequent difficulty in identifying sufficient negative examples within mini-batches.
The introduced framework exhibits versatility, being adaptable to current encoder-based methodologies.

The proposed framework is built around a specific setting (3-task Chinese LJP), and there is no evidence that a similar methodology would be useful in other related LJP tasks from other jurisdictions (Chalkidis et al., 2019; Niklaus et al., 2021; and others).